META-RESEARCH ARTICLE

# Breaking the reproducibility barrier with standardized protocols for plant–microbiome research

Vlastimil Novak[1], Peter F. Andeer[1], Eoghan King[2,3], Jacob Calabria[4],
Connor Fitzpatrick[5], Jana M. Kelm[6], Kathrin Wippel[7,8], Suzanne M. Kosina[1],
Benjamin P. Bowen[1,2], Chris Daum[2], Matthew Zane[2], Archana Yadav[9], Mingfei Chen[9],
Dor Russ[5], Catharine A. Adams[1,10], Trenton K. Owens[1], Bradie Lee[2,10], Yezhang Ding[1],
Zineb Sordo[11], Romy Chakraborty[9], Simon Roux[2], Adam M. Deutschbauer[1,10],
Daniela Ushizima[11], Karsten Zengler[12,13,14,15], Borjana Arsova[6], Jeffery L. Dangl[5],
Paul Schulze-Lefert[7], Michelle Watt[4], John P. Vogel[2]*, Trent R. Northen[1,2]*

**1** Environmental Genomics and Systems Biology, Lawrence Berkeley National Laboratory, Berkeley, California, United States of America, **2** The DOE Joint Genome Institute, Lawrence Berkeley National Laboratory, Berkeley, California, United States of America, **3** Centro de Biotecnología y Genómica de Plantas, Universidad Politécnica de Madrid (UPM)–Instituto Nacional de Investigación y Tecnología Agraria y Alimentación (INIA/CSIC), Campus de Montegancedo, Madrid, Spain, **4** Faculty of Science, School of BioSciences, The University of Melbourne, Parkville, Australia, **5** Department of Biology and Howard Hughes Medical Institute, University of North Carolina at Chapel Hill, Chapel Hill, North Carolina, United States of America, **6** Institute for Bio- and Geosciences, Plant Sciences (IBG-2), Forschungszentrum Jülich GmbH, Jülich, Germany, **7** Max Planck Institute for Plant Breeding Research, Cologne, Germany, **8** Swammerdam Institute for Life Sciences, University of Amsterdam, Science Park Amsterdam, Amsterdam, The Netherlands, **9** Earth and Environmental Sciences, Lawrence Berkeley National Laboratory, Berkeley, California, United States of America, **10** Department of Plant and Microbial Biology, University of California-Berkeley, Berkeley, California, United States of America, **11** Computing Sciences Area, Lawrence Berkeley National Laboratory, Berkeley, California, United States of America, **12** Department of Pediatrics, University of California, San Diego, La Jolla, California, United States of America, **13** Department of Bioengineering, University of California, San Diego, La Jolla, California, United States of America, **14** Center for Microbiome Innovation, University of California, San Diego, La Jolla, California, United States of America, **15** Program in Materials Science and Engineering, University of California, San Diego, La Jolla, California, United States of America

* jpvogel@lbl.gov (JPV); trnorthen@lbl.gov (TRN)

## Abstract

Inter-laboratory replicability is crucial yet challenging in microbiome research. Leveraging microbiomes to promote soil health and plant growth requires understanding underlying molecular mechanisms using reproducible experimental systems. In a global collaborative effort involving five laboratories, we aimed to help advance reproducibility in microbiome studies by testing our ability to replicate synthetic community assembly experiments. Our study compared fabricated ecosystems constructed using two different synthetic bacterial communities, the model grass *Brachypodium distachyon*, and sterile EcoFAB 2.0 devices. All participating laboratories observed consistent inoculum-dependent changes in plant phenotype, root exudate composition, and final bacterial community structure, where *Paraburkholderia* sp. OAS925 could dramatically shift microbiome composition. Comparative genomics and exudate

**Data availability statement:** All of the links to the protocols and data as well as the study metadata are available via NMDC at https://data.microbiomedata.org/details/study/nmdc:sty-11-ev70y104. The 16S rRNA amplicon sequencing data are available via NCBI (https://www.ncbi.nlm.nih.gov/) as BioProject PRJNA1151037. All raw data, including plant phenotypes, sterility tests, metabolite identifications, in vitro assays, and code are available via Figshare at https://doi.org/10.6084/m9.figshare.c.7373842. The untargeted metabolomics outputs (HILIC-pos) with features annotations and .mzml files are available via GNPS2 at https://gnps2.org/status?task=2ccbf82840724c99a2ac-c2c9e512a302. Raw LC–MS/MS files are available in .raw format at MassIVE (https://massive.ucsd.edu/) under ID number MSV000095476 or via https://doi.org/10.25345/C5Q23RB6B. The protocol is available at protocols.io via https://dx.doi.org/10.17504/protocols.io.kx-ygxyydkl8j/v1. The annotated bacterial genomes can be accessed via IMG/M (https://img.jgi.doe.gov/) by searching for either the isolate name, taxon ID, or GOLD Project ID; on Hugging Face (https://doi.org/10.57967/hf/5885); or by using the links in S2 Table.

**Funding:** We gratefully acknowledge the financial support from the U.S. Department of Energy (DOE) Office of Science, Office of Biological and Environmental Research for support of the Trial Ecosystem Advancement for Microbiome Science (TEAMS) led by LBNL, Award DE-SC0021234 led by UC San Diego, and m-CAFEs Microbial Community Analysis & Functional Evaluation in Soils (m-CAFEs@lbl.gov), a Science Focus Area led by Lawrence Berkeley National Laboratory where all of the LBNL led projects are under Contract No. DE-AC02-05CH11231. J.C. and M.W. acknowledge support from the University of Melbourne Botany Foundation. The funders had no role in study design, data collection and analysis, decision to publish, or preparation of the manuscript.

**Competing interests:** I have read the journal's policy and the authors of this manuscript have the following competing interests: P.F.A. and T.R.N. are inventors of patent US11510376B2, held by the University of California, covering an Ecosystem device for determining plant-microbe interactions. All other authors have declared that no competing interests exist.

utilization linked the pH-dependent colonization ability of *Paraburkholderia*, which was further confirmed with motility assays. The study provides detailed protocols, benchmarking datasets, and best practices to help advance replicable science and inform future multi-laboratory reproducibility studies.

## 1. Introduction

As recent perspective papers have highlighted, establishing model microbiomes is a pressing need in environmental microbiology [1,2]. Several years ago, a vision was presented for developing and validating standardized 'fabricated ecosystems' to enable replicable studies of microbiomes in ecologically relevant contexts, akin to the adoption of shared model organisms [1]. A fabricated ecosystem is defined as a closed laboratory ecological system where all biotic and abiotic factors are initially specified/controlled. Synthetic microbial communities (SynComs) are valuable tools for bridging the gap between natural communities and studies involving axenic cultures and isolates [3]. By limiting complexity yet retaining functional diversity and microbe-microbe interactions, SynComs can be used to unravel mechanisms underlying complex interactions, providing critical insights into community assembly processes, microbial interactions, and host physiology, e.g., plant host [3–6]. These interactions between the host and its microbes define the holobiont concept, where the plant and its microbiome form a single dynamic ecological unit [7]. However, standardization is essential to fully leverage the potential of SynComs and achieve replicable plant microbiome studies [8]. This requires overcoming several challenges, including the availability of strains and standardized protocols for their growth in the laboratory. To address these challenges, we recently developed a standardized model community of 17 bacterial isolates from a grass rhizosphere available through a public biobank (DSMZ), along with cryopreservation and resuscitation protocols [9].

Other aspects to enable replicable microbiome studies must be standardized, including sterile habitats and protocols for sample collection and analysis [1]. As initial steps towards this vision, we developed a first-generation sterile container for fabricated ecosystems (EcoFAB device) and performed a multi-laboratory study demonstrating the reproducible physiology of the model grass *Brachypodium distachyon* [10]. Recently, it was found that *Paraburkholderia* sp. OAS925 dominated other members of the model 17-member SynCom for *B. distachyon* root colonization [11]. Additionally, we have since developed an improved EcoFAB 2.0 device that enables highly reproducible plant growth [12]. The next step towards standardization is to test the replicability of microbiome formation, plant responses to microbiomes, and root exudation using these standardized laboratory habitats and SynComs. This can be achieved through inter-laboratory comparison studies or ring trials—a powerful tool in proficiency testing of analytical methods [13,14] that are currently underutilized in microbiome research.

Here, we describe a five-laboratory international ring trial investigating the reproducibility of *B. distachyon* phenotypes, exometabolite profiles, and microbiome

assembly within the EcoFAB 2.0 device. The experiment compared the recruitment of the full SynCom versus one lacking the dominant root colonizer *Paraburkholderia* sp. OAS925 [11]. To minimize variation required in all laboratories, almost all supplies (e.g., EcoFABs 2.0, seeds, SynCom inoculum, filters) were distributed from the organizing laboratory, and detailed protocols, including annotated videos, were created. Each laboratory measured plant phenotypes and collected samples for 16S rRNA amplicon sequencing and metabolomic analyses by LC–MS/MS. A single laboratory performed all the sequencing and metabolomic analyses to minimize analytical variation. Follow-up in vitro assays and comparative genomics were conducted to gain insights into mechanisms leading to *Paraburkholderia* sp. OAS925 dominance. Overall, the study demonstrates consistent plant traits across multiple laboratories and provides publicly accessible benchmarking data for other labs to leverage, replicate, and extend this work. In addition, we describe the challenges we encountered in performing this study, thus providing information that can facilitate future microbiome reproducibility studies.

**Abbreviations :** ANOVA, analysis of variation; CFU, colony-forming unit; CV, coefficient of variation; DAI, days after inoculation; DAP, diaminopimelic acid; DSMZ, Leibniz-Institute DSMZ-German Collection of Microorganisms and Cell Cultures; EcoFAB, fabricated ecosystem ; IMG/M, Integrated Microbial Genomes and Microbiomes; KEGG, Kyoto Encyclopedia of Genes and Genomes; LB, Luria–Bertani; LBNL, Lawrence Berkeley National Laboratory; LC-MS/MS, liquid chromatography-tandem mass spectrometry; MS, Murashige and Skoog; NMDC, National Microbiome Data Collaborative; NMDS, Non-metric Multidimensional Scaling; OD600, optical density at 600 nm; OTU, operational taxonomic unit; RFP, red fluorescent protein; RT, retention time; SynCom, synthetic community; T3SS, Type 3 Secretion System.

## 2. Results

### 2.1. Study design and logistics

Our main objective was to develop and test methods to reproducibly study plant microbiomes within the sterile EcoFAB 2.0 device (Fig 1a). We hypothesized that the inclusion of *Paraburkholderia* sp. OAS925, a dominant *B. distachyon* root colonizer into SynCom [11], would reproducibly influence the microbiome assembly, metabolite production, and plant growth across multiple laboratories using the EcoFAB 2.0 device. To test the hypothesis, we deployed the grass *B. distachyon* with a SynCom consisting of 16 or 17 members (either with or without *Paraburkholderia* sp. OAS925) that was originally developed to span the diversity of bacteria isolated from a grass rhizosphere, including representatives from the Actinomycetota, Bacillota, Pseudomonadota, and Bacteroidota phyla (Fig 1b) [9]. Our study was conducted across five laboratories (designated A–E) and consisted of four treatments with seven biological replicates each (Fig 1a): an axenic (mock-inoculated) sterile plant control, SynCom16-inoculated plants, SynCom17-inoculated plants, and plant-free medium control. Each laboratory followed written protocols and annotated videos, gathered root and unfiltered media samples for 16S rRNA amplicon sequencing, filtered media for metabolomics, measured plant biomass, and performed root scans. At the end of the study, the collected data and samples were sent to the organizing laboratory for sequencing, metabolomics, and data analysis.

The detailed protocol with embedded annotated videos used by all five laboratories is available via protocols.io (https://dx.doi.org/10.17504/protocols.io.kxygxyydkl8j/v1) [15]. The general procedure follows these steps: (i) EcoFAB 2.0 device assembly; (ii) *B. distachyon* seed dehusking, surface sterilization, and stratification at 4 °C for 3 days; (iii) Germination on agar plates for 3 days; (iv) Transfer of seedlings to the EcoFAB 2.0 device for an additional 4 days of growth; (v) Sterility test and SynCom inoculation into the EcoFAB 2.0 device; (vi) Water refill and root imaging at three timepoints; (vii) Sampling and plant harvest at 22 days after inoculation (DAI). Since

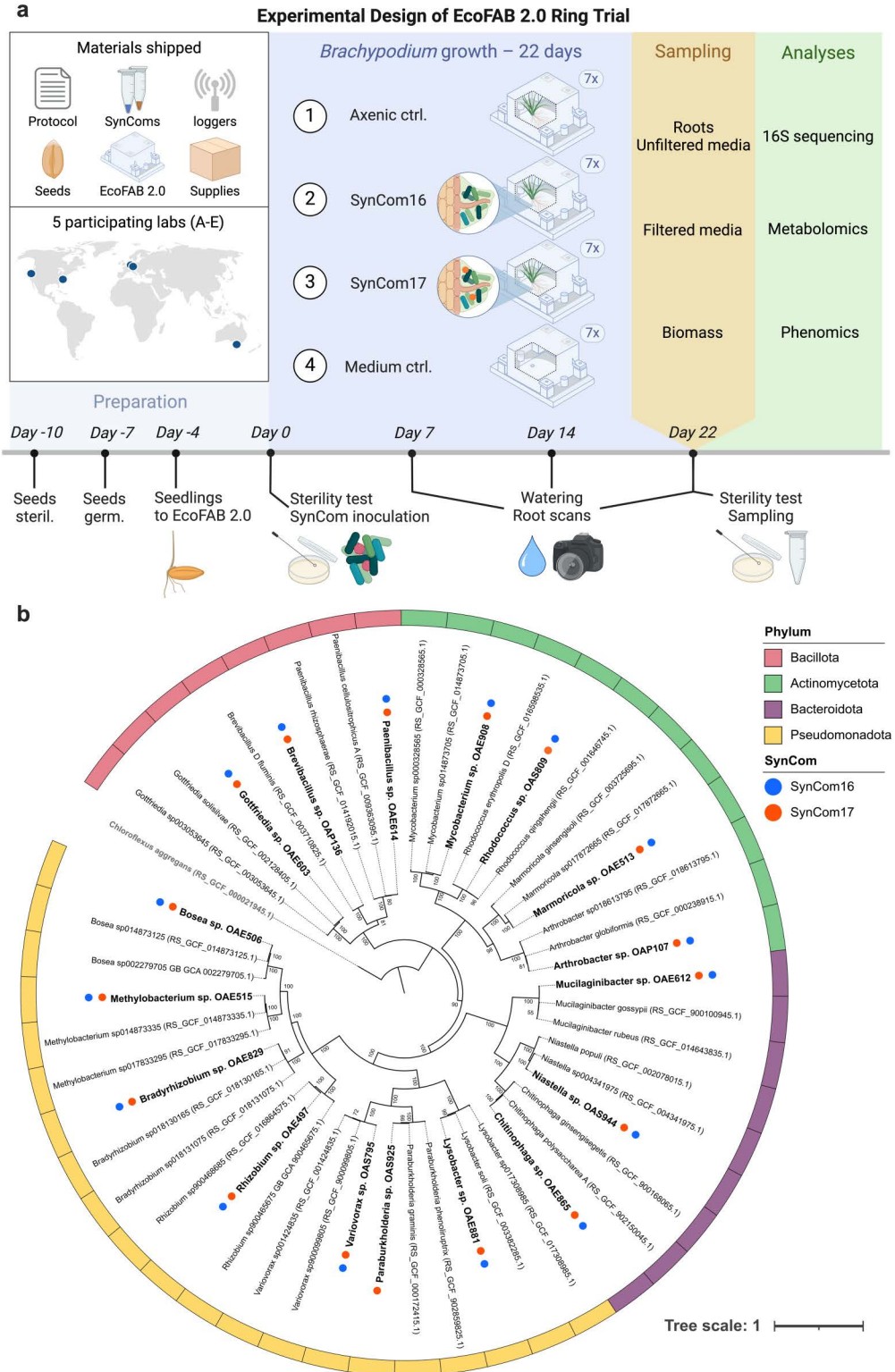

**Fig 1. Overview of the interlaboratory comparison study. (a)** Experimental design where five laboratories across three continents conducted the same experiment using shipped materials. These included a detailed protocol (https://dx.doi.org/10.17504/protocols.io.kxygxyydkl8j/v1) [15], SynComs and Mock solution stocks, light and temperature loggers, *Brachypodium distachyon* seeds, EcoFAB 2.0 device parts, and various lab supplies (growth

medium, filters, sampling tubes). We inoculated *B. distachyon* plants with either a 16- or 17-member SynCom, with controls being axenic plants and medium-only technical control (Mock-inoculated), *n* = 7. We tested sterility and imaged roots at multiple time points. Finally, we quantified plant biomass, analyzed exudate metabolite composition, and measured root and medium microbiomes. **(b)** The phylogenomic tree is based on 120 marker genes, where SynCom members are highlighted in bold, with phylum-level classification shown by colored strips and SynCom membership by circles (Syn-Com16 in blue, SynCom17 in orange). The 2 closest taxonomic genomes are included, with GenBank accession numbers in parentheses. Nodes with over 50 bootstrap support values from 100 replicates are labeled. *Chloroflexus aggregans* (bold gray) served as an outgroup. The 17 genomes can be accessed via Hugging Face (https://doi.org/10.57967/hf/5885) [16] or links in S2 Table.

differences in labware and material can cause experimental variation, the protocol specifies the part numbers used in this study. Organizers provided critical components, including growth chamber data loggers, in the initial package of nonperishable supplies, while the SynComs and freshly collected seeds were shipped just before the study. Given the time zone differences, it was difficult to synchronize all activities, so each laboratory performed the experiment independently within 1.5 months of each other (S1 Table). All participants followed data collection templates and image examples.

## 2.2. Protocols resulted in reproducibly sterile conditions

During the study, all participating laboratories tested the sterility of the EcoFABs 2.0 devices by incubating spent medium on Luria–Bertani (LB) agar plates at two different time points. Less than 1% (2 out of 210) of all tests showed colony formation (Fig 2a). Namely, a single colony was observed in one treatment of laboratory D in SynCom17, and multiple colonies for laboratory B in medium-only control (plate had cracked lid).

## 2.3. Reproducible plant growth

When plant biomass data were combined across laboratories, we observed a significant decrease in shoot fresh weight and dry weight of plants inoculated with SynCom17, and to a lesser extent SynCom16, relative to the axenic treatment (Fig 2b). This said, we did observe some variability between laboratories (S1 Fig), which is presumably due to growth chamber differences including light quality (fluorescent versus LED growth lights), light intensity and temperature (S1 Table). Supporting this, the data loggers revealed variability in measured temperatures (S2a Fig) and photoperiod (S2b Fig). Image analysis of scanned roots (S3 Fig) revealed that SynCom17 caused a consistent decrease in root development observed after 14 DAI onwards (Fig 2c).

## 2.4. Reproducible microbiome assembly

SynComs were prepared using optical density at 600 nm ($OD_{600}$) to colony-forming unit (CFU) conversions (S2 Table) to ensure equal cell numbers (final inoculum $1 \times 10^5$ bacterial cells per plant) and shipped on dry ice to each laboratory as 100× concentrated stocks in 20% glycerol. The cells were resuspended and added to 10-day-old *B. distachyon* seedlings in the EcoFAB 2.0. After 22 days of growth, the roots and media were sampled, shipped back to the organizing laboratory, sequenced (see S3 Table for read counts), and compared to the original inoculum. For both SynComs (SynCom16 and SynCom17), the community composition at 22 DAI differed from the inoculum (Fig 3). As hypothesized, the root microbiome inoculated with SynCom17 was dominated by *Paraburkholderia* sp. OAS925 across all laboratories (98±0.03% average relative abundance±SD). In its absence (SynCom16), other isolates showed high relative abundance in the root microbiome with increased variability across laboratories, namely *Rhodococcus* sp. OAS809 (68±33%), *Mycobacterium* sp. OAE908 (14±27%), and *Methylobacterium* sp. OAE515 (15±20%). The most dominant microbial isolates detected in root samples were also typically present in the media samples (S4a Fig). Ordination plots showed clear separations between SynCom16 and SynCom17 microbiomes for both root and media, with generally higher variability between samples for the SynCom16 microbiome (S4b Fig). There was a minimal contribution of unknown reads in all samples, consistent with the observed sterility of the controls. The three samples with the highest proportion of unknown reads (>2.5%)

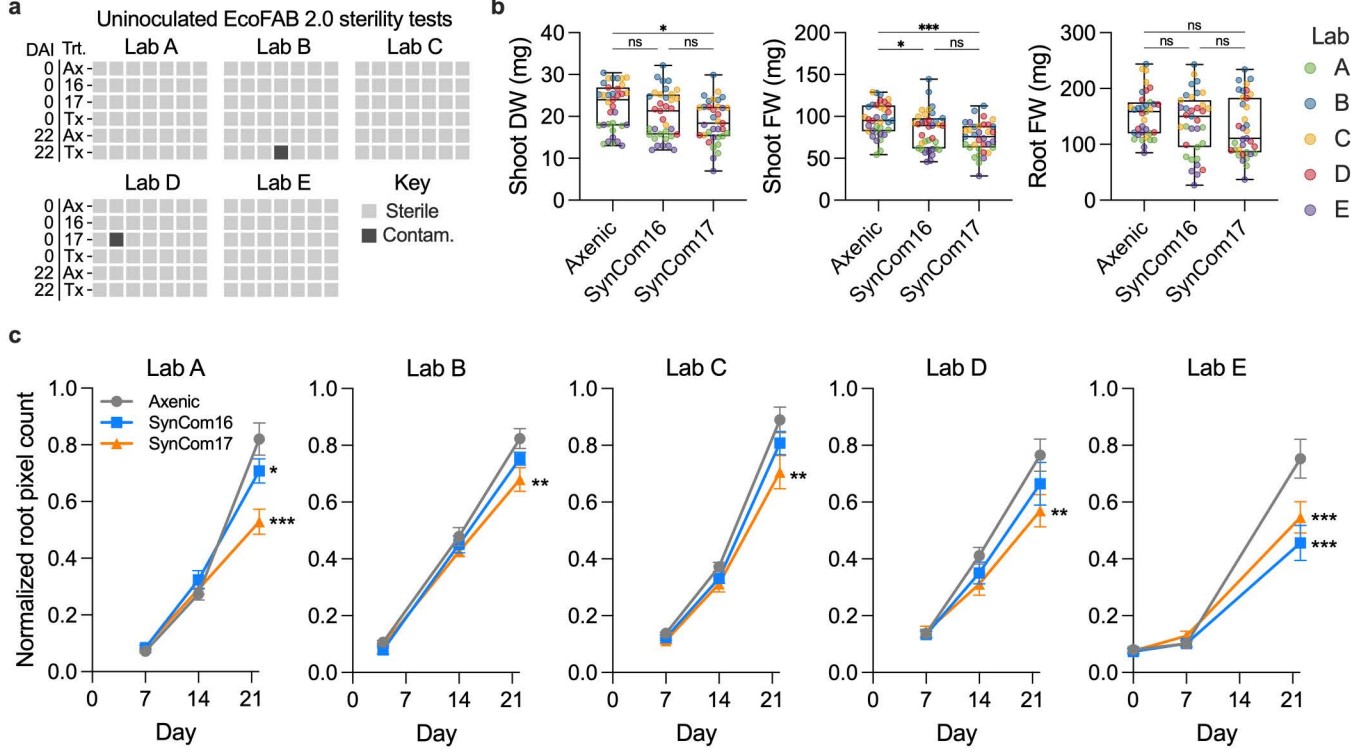

**Fig 2. Plant phenomics and EcoFAB 2.0 device sterility.** (a) Sterility of uninoculated EcoFAB 2.0 devices tested across laboratories A–E and treatments (Trt.: Ax-Axenic, 16-SynCom16, 17-SynCom17, Tx-Medium technical control) at day 0 and day 22 after inoculation (DAI). The medium from these devices was incubated on LB agar plates for 22 days to observe bacterial colony formation. Each square represents one test with gray fill indicating sterility and black contamination. The photos of the agar sterility test can be found at https://doi.org/10.6084/m9.figshare.26409220. (b) Plant biomass weight combined across all laboratories (Lab A–E in different colors), measured as shoot dry weight, shoot and root fresh weight. One-way ANOVA with Tukey test, $n = 7$, ns $p > 0.5$, $*p < 0.05$, $**p < 0.01$, $***p < 0.001$. The shoot photos can be found at https://doi.org/10.6084/m9.figshare.26409310. (c) Root system development was analyzed using RhizoNet (Lab B–E) and ImageJ (Lab A). The raw root pixel counts were normalized to the maximum value in each lab. Two-way ANOVA with Dunnet's test vs. Axenic control, $n = 7$, ns $p > 0.5$, $*p < 0.05$, $**p < 0.01$, $***p < 0.001$. The root scans and Rhizonet reports can be found at https://doi.org/10.6084/m9.figshare.26131291. The data underlying this figure can be found at https://doi.org/10.6084/m9.figshare.26401315.

contained operational taxonomic units (OTUs) associated with human microbes (S4 Table), suggesting introduction during handling. Furthermore, SynCom17 treatment in laboratory D did not show unknown reads (S3 Table), suggesting that the failed sterility test (Fig 2a) was likely caused by plate downstream contamination.

## 2.5. Reproducible rhizosphere metabolome

The spent medium from each fabricated ecosystem was filtered and shipped to the organizing laboratory for LC–MS/MS analysis (S5 Table), followed by targeted and untargeted metabolomics to determine the root exudate composition and metabolite profiles in the presence of different SynComs in the rhizosphere. The targeted analysis identified 60 metabolites spanning diverse metabolite classes (S6 Table). Hierarchical clustering revealed general clustering by treatment and not laboratory (Fig 4), consistent with the experimental reproducibility observed with plant growth phenotypes and root microbiome composition. Furthermore, the metabolite clustering showed several treatment-dependent metabolite changes. The first large cluster included diverse metabolites increased in the SynCom17 treatment. A second large cluster consisted of metabolites with lower relative concentrations in the SynCom17 treatment, represented mainly by amino

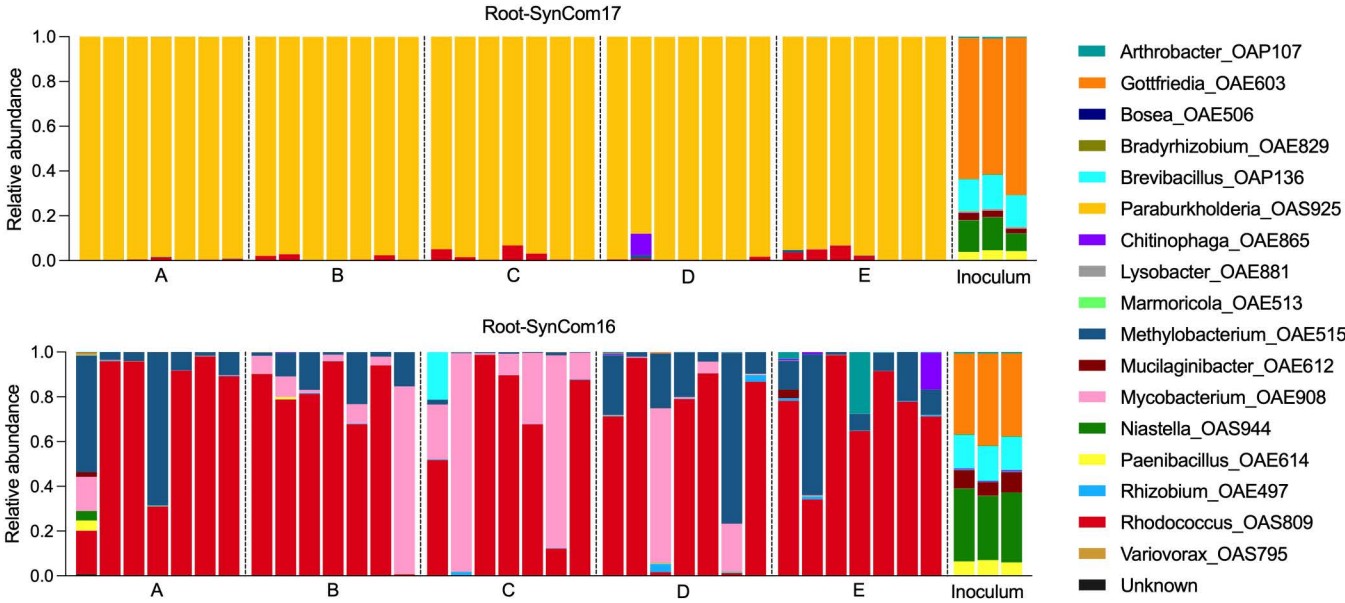

**Fig 3. Root microbiome.** Microbiome composition of *Brachypodium distachyon* roots and starting inoculum of plants inoculated with SynCom16 or SynCom17 (±*Paraburkholderia* sp. OAS925). Letters indicate different laboratories, with each biological replicate shown (*n* = 7). The inoculum shows technical replicates (*n* = 3). The data underlying this figure can be found at https://doi.org/10.6084/m9.figshare.26401315 or in S3 Table.

acids. A third, much smaller cluster consisting primarily of nucleosides(tides) increased in the SynCom16 or both SynCom treatments. This finding highlights the prominent impact of the community dominated by *Paraburkholderia* sp. OAS925 on modulation of metabolite composition in the rhizosphere. This was further supported by untargeted metabolomics on 833 detected features that showed a clear separation between rhizosphere metabolomes of axenic plants and SynCom17, which was reproducible across all laboratories (S5 Fig). These changes may be due to metabolite production or uptake by the microbes or plant roots or the activity of extracellular enzymes [17–19].

## 2.6. Statistical evaluation of variation between labs

Ordination analysis using non-metric multidimensional scaling (NMDS) for microbiome (S4b Fig) and metabolome data (S5 Fig) showed a clear separation between treatments, but some separation between labs. To explore this in more detail, we quantified the lab- versus treatment-associated variation by analysis of variation (ANOVA) (S7 Table), pairwise tests (S8 Table), and coefficient of variation (CV).

ANOVA analysis of microbial abundance showed that more microbes were significantly different due to the treatments versus lab doing the experiment, across both roots and media (S6a Fig). When investigating the source of variation, we observed microbe-dependent results, with the lab accounting for <18% of the variation, while SynCom treatment had a larger contribution in several microbes, especially *Praburkholderia*, *Rhodococcus*, and *Methylobacterium* (S6b Fig). CV analysis showed consistent trends for both root and media samples, with SynCom16 displaying higher within-lab variability than SynCom17 in all labs (S6c Fig). Between-lab CVs were similar across treatments. Overall, the observed microbial abundance levels were more dependent on treatment than lab.

Metabolite levels were more affected by the lab-associated factors than the microbial abundances. However, treatment was still the primary influence in the majority of the metabolites. Both lab and treatment significantly affected the majority of the 60 identified metabolites (S7a Fig). Comparing these results between the various labs, we found that 7–23

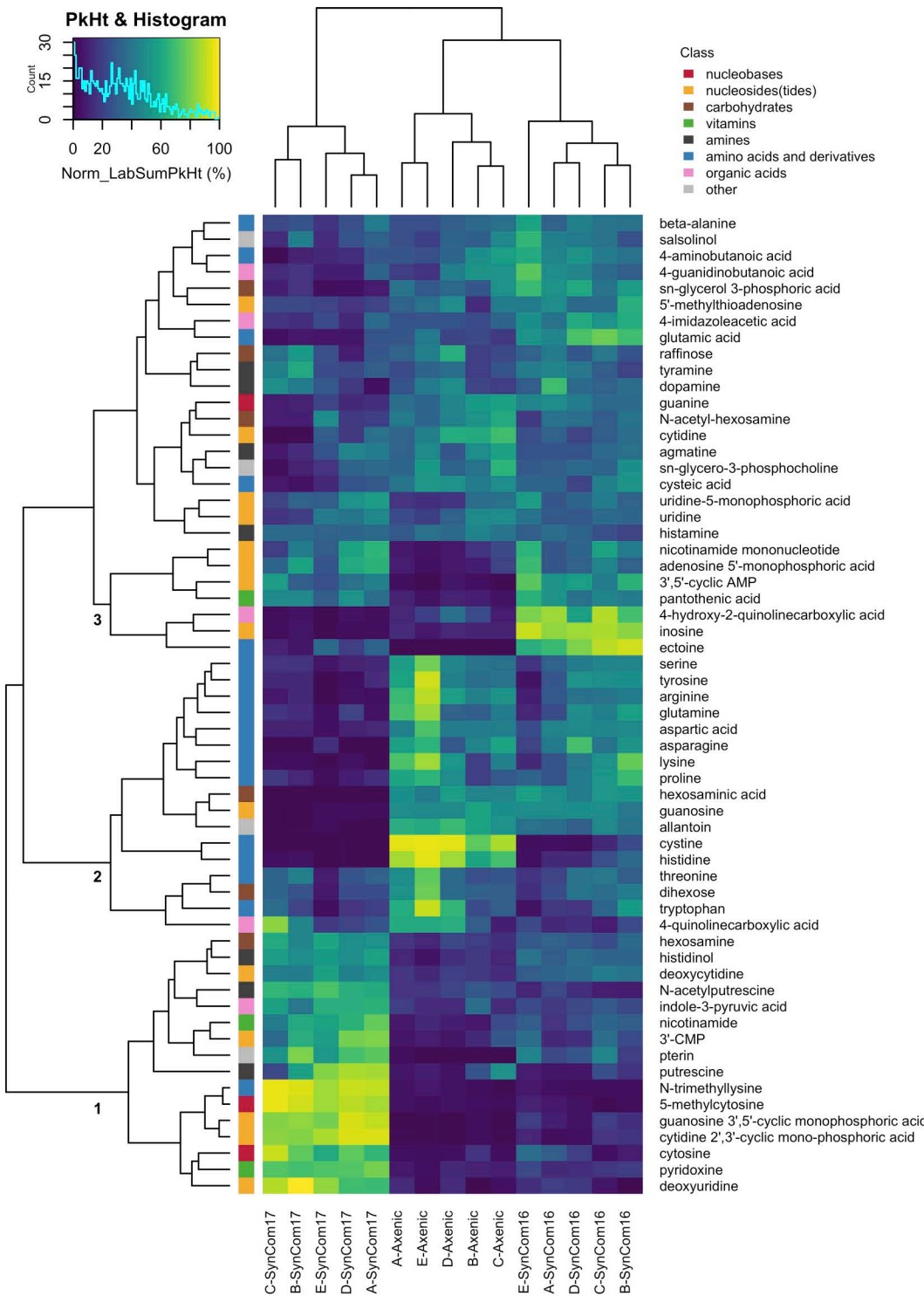

**Fig 4. Targeted metabolomic analysis of rhizosphere.** We show mean values for 60 identified metabolites for each lab/treatment combination (*n* = 7), row-normalized to the average sum peak height per lab. Row colors indicate metabolite classes. Cluster 1: Abundant in SynCom17; Cluster 2: Low in SynCom17; Cluster 3: Abundant in SynCom16 or both SynComs. The data underlying this figure can be found at https://doi.org/10.6084/m9.figshare.26401315 or in S6 Table.

metabolites were significantly different in specific pairwise lab comparisons, with lab pair A–B being the most different, and labs A–D, C–D, and B–D most similar to one another (S7b Fig). This said, most metabolites were primarily influenced by treatment rather than lab, though a subset, especially salsolinol, tyramine, and dopamine showed notable lab effects (S7c Fig). Metabolites in Axenic and SynCom17 treatments had lower CV than SynCom16 in all labs, and the within-lab CV was approximately the same as the between-lab CV (S7d Fig). Together, these results support the conclusion that metabolite abundances were largely treatment-driven, though some metabolites were sensitive to lab-specific differences.

### 2.7. Colonization by *Paraburkholderia*

Given the reproducible impacts of *Paraburkholderia* sp. OAS925 on our fabricated ecosystems, including plant growth phenotypes, microbiome structure, and rhizosphere exometabolites, we performed additional analyses to gain insights into potential mechanisms that may explain its dominance. Comparative genomic analysis showed that the *Paraburkholderia* sp. OAS925 genome (IMG/M Taxon ID: 2931840637) uniquely includes acid resistance genes such as glutamate and arginine transporters and decarboxylases (S8 Fig) and a gene module coding for a Type 3 Secretion System (T3SS), which was not found in any other member of the SynCom (S9 Table).

We inoculated *B. distachyon* with a red fluorescent protein (RFP) expressing *Paraburkholderia* sp. OAS925 to investigate spatial-temporal root colonization in EcoFAB 2.0. Clear RFP signals were detected at the root tip and in the maturation zone at 1 DAI, with increased biofilm formation observed at 3 DAI (S9 Fig). We noted both sessile colonies on the rhizoplane (S1 Video) and active swimming surrounding root cells (S2 Video).

The biofilm formation and motility observed during microscopy motivated the follow-up in vitro assays to further assess these characteristics across isolates. *Paraburkholderia* sp. OAS925 exhibited the sixth-highest biofilm formation and the most liquid-culture growth (S10 Fig), highlighting its potential to outgrow many other bacteria in SynCom17 [20].

Swimming motility assays on soft agar revealed that *Paraburkholderia* sp. OAS925 had the highest motility within the first 24 h (S11a Fig). Additionally, compared to other isolates with similar motility phenotypes (S11b Fig), it maintained fast swimming in acidic conditions (S11c Fig) in the range of the hydroponic medium (pH 5.5–6.0 at the start of the experiment). KEGG mapping (map02040) showed the presence of flagellar assembly genes, suggesting that the observed motility is due to flagella.

## 3. Discussion

There is an urgent need to move towards replicable experimental systems to address common difficulties in reproducing microbiome experiments [2,21]. Here, we report what, to our knowledge, is the first multi-laboratory microbiome reproducibility study. We constructed fabricated ecosystems using two SynComs, the model plant *B. distachyon* Bd21-3, and sterile EcoFAB 2.0 devices. These, in combination with written protocols and annotated videos, generally resulted in reproducible plant growth phenotypes, host microbiomes, and exometabolomes across five laboratories spanning three continents. Of the 210 sterility tests performed across the 5 labs, only 2 potential contamination events were detected (**Fig 2a**). One of these was from a plate with a cracked lid, and so we conclude that the EcoFAB 2.0 devices and protocols were effectively able to achieve sterility across all labs.

The magnitude of laboratory-specific effects on microbiome and metabolome data varied across our study, with laboratory factors explaining <18% of the variation in 16S rRNA sequencing data, but having a greater impact on specific exometabolites, where 15 metabolites had variation >18% (S7 Table). For example, hexosaminic acid, deoxycytidine, and N-acetylputrescine were highly reproducible between labs, whereas salsolinol, tyramine, and dopamine showed greater variability. The biochemical connection between these metabolites (salsolinol derived from dopamine, which is synthesized from tyramine [22]) and dopamine's instability (prone to oxidation [23]) may contribute to their high variability across laboratories. These findings suggest opportunities to reduce metabolite variability through refined sample collection, handling, and storage. Together, these findings emphasize the importance of accounting for laboratory-specific effects

when interpreting metabolome data, and highlight the need for standardized methods to minimize variability to generate high-quality data that can be integrated across labs (e.g., using AI).

Our study confirms previous findings that *Paraburkholderia* sp. OAS925 dominates the root microbiome of *B. distachyon* when part of the SynCom [11]. Similar results were observed for another grass, *Avena barbata*, grown in its native soil, where members of the order Burkholderiales were the most active bacteria in the rhizosphere based on carbohydrate depolymerization [24]. Soil pH, organic carbon availability, oxygen levels and redox status are key factors influencing microbial community composition [25]. Our study suggests several traits that likely help contribute to the dominance of *Paraburkholderia* sp. OAS925 in SynCom17. For example, its high motility in acidic environments (S11c Fig), such as the rhizosphere, might facilitate quick colonization of ecological niches and affect community assembly [26–28]. Its ability to utilize amino acids like arginine, glutamine, and glutamate (Fig 4) provides a possible mechanism for cytoplasm de-acidification to maintain motility-enabling transmembrane proton gradient [29,30]. These results align with a previous study showing that *Pseudomonas simiae* genes involved in motility, carbohydrate metabolism, cell wall biosynthesis, and amino acid transport aid in colonization of *Arabidopsis* roots [31]. Interestingly, in SynCom16, *Rhodococcus* often dominates on roots (Fig 3) and shares fast growth (S10 Fig) and high motility (S11a Fig) with *Paraburkholderia*.

Both SynCom16 and SynCom17 decreased plant biomass, with SynCom17 having a more pronounced effect (Figs 2 and S1). The observed dominant colonization by *Paraburkholderia* (S9 Fig, S1 and S2 Videos) or other bacteria might disrupt plant nutrient homeostasis, as the root microbiome plays a crucial role in forming root diffusion barriers and maintaining plant mineral nutrient balance [5]. Furthermore, the observation of a T3SS (S9 Table) is consistent with previous findings in *Paraburkholderia* genomes and has been shown to play a role in root colonization and virulence. Future studies should investigate the role of T3SS in the dominance of *Paraburkholderia* sp. OAS925 in SynCom17 treatments and the associated plant biomass decrease. Additionally, future testing of the effects of *Paraburkholderia* sp. OAS925 and other dominant isolates on plant growth can provide additional insights into the extent that the observed growth effects are indeed a direct result of a single strain.

By organizing this ring trial, we learned valuable lessons that can be useful for future studies (Fig 5). First, it is important to perform pilot studies to optimize methods before initiating any multi-laboratory study. Long-distance sample and inoculum shipping posed challenges, especially given unpredictable long delays in customs and potential thawing due to dry ice sublimation [32]. Microorganism shipments require engagement with shippers and familiarity with country-specific import/export legal regulations. We also observed variability in plant biomass (Fig 2b), which could be attributed to differences, especially in light and temperature (S2 Fig) between the growth chambers used in each laboratory (S1 Table). Additionally, standardization of the scanner model for automated root system analysis could help reduce variability and improve reproducibility. Ideally, the same equipment would be used, with a real-time readout of environmental conditions, although this would significantly increase the cost of the study.

Despite our detailed protocols and annotated videos, several challenges remain to replicate microbiome studies, underscoring the importance of using the data from this study to benchmark future studies. We recommend using the comment section on protocols.io for ongoing refinement and clarification, allowing the procedure to evolve as a living document [33]. To provide FAIR (Findable, Accessible, Interoperable, and Reusable) data access and enable others to use these data for benchmarking, integration, and extension, all data from this study are available via the National Microbiome Data Collaborative (NMDC) project page (https://data.microbiomedata.org/details/study/nmdc:sty-11-ev70y104) [34]. The EcoFAB 2.0 devices (up to 50 pieces), *B. distachyon* Bd21-3 plant line, and metabolomics methods used in this study are currently freely available to domestic and international researchers via Joint Genome Institute (JGI) User Programs (https://jgi.doe.gov/), subject to a competitive scientific review process. The 16S rRNA sequencing is readily available via commercial and academic sequencing centers. Although the relative abundance of organisms should ideally not correlate with the sequencing facility, sample handling, DNA extraction, and bioinformatics can significantly impact results, underscoring the

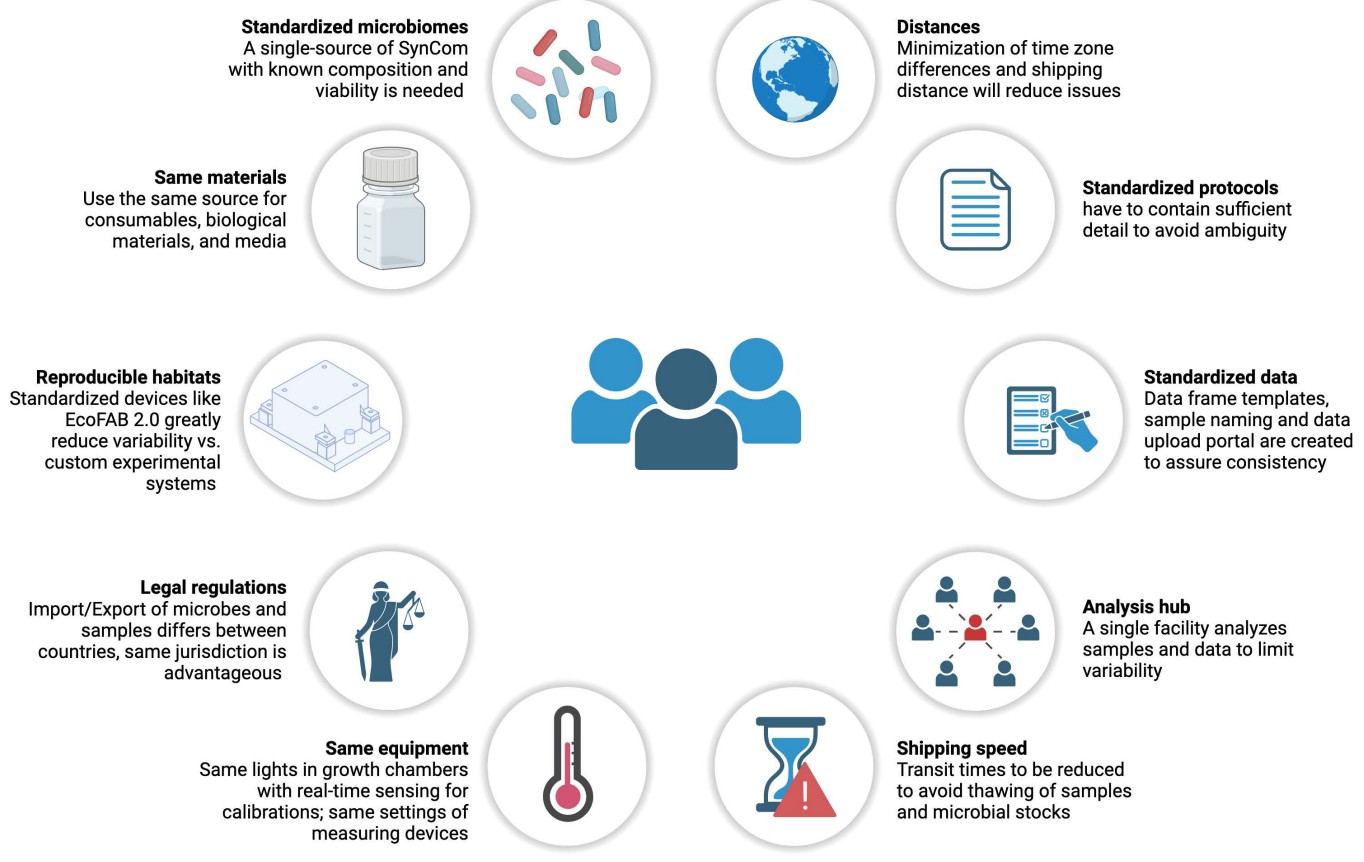

**Fig 5. Framework for conducting reproducible multi-lab microbiome studies.** This figure summarizes key factors and lessons for standardizing microbiome experiments. It highlights the essential elements for organizing reproducible studies and offers insights into the organization of future micro-biome multi-lab studies.

need to consider protocols when making comparisons [35]. Another challenge we see is that the strains of our SynCom are currently available as individual strains, so batch variation would be reduced if culture collections or private companies provided ready-to-use SynCom mixtures.

This study demonstrates that multiple geographically dispersed laboratories can reproduce SynCom-driven changes in plant phenotypes, community assembly, and exometabolite profiles. This was a challenging yet essential step in the vision outlined by Zengler and colleagues [1] to verify the reproducibility of experimental systems and protocols, which enable scientists to replicate and build on each other's work. We see several ways these methods can help advance the field: first, scientists can replicate the study and compare their results against those reported here before extending the findings with additional modifications (e.g., adding phages, fungi, engineered strains, different hosts, new devices, etc.). Second, scientists can generate experimental data through replication and benchmarking, enabling integrative computational analyses that control laboratory-specific effects. Providing FAIR data and accompanying metadata and protocols, as done here, will be an essential step in achieving this vision. Such efforts would greatly enhance the application of machine learning to make generalizable discoveries drawn from multiple studies, ultimately leading to understanding microbial processes in complex natural environments.

## 4. Materials and methods

### 4.1. Preparation of synthetic bacterial communities

The bacterial isolates originate from the switchgrass rhizosphere and are available at DSMZ as individual strains [9] or as a collection (DSM 200000SY, inquire at contact@dsmz.de). Glycerol stocks had their 16S gene (27F - 1492R) sequenced to confirm isolate identity and purity. Each verified isolate was streaked on a preferred medium (S2 Table), and a single colony was inoculated into liquid culture. After 2–8 days at 27 °C, cultures were centrifuged at 5,000 g and washed with ½ Murashige and Skoog (MS) basal salts medium. The washed cultures were used to create SynComs by measuring $OD_{600}$, converting to CFU, and combining them to 2e7 cells/ml each (1:1 ratio) in 20% glycerol for cryopreservation [9]. We used SynComs composed of equal numbers of cells for the constituent strains, which is a widely used approach [9,36,37]. A 20% glycerol in ½ MS basal salts was a mock solution. Solutions were aliquoted (100 µL) in microcentrifuge tubes and stored at −80 °C. The participants diluted the solutions 100-fold, assuming 50% cell survival during freezing-thawing, resulting in a final theoretical CFU of 1e5/plant. Serial dilution established the $OD_{600}$ to CFU conversions, and a hemocytometer was used to measure *Gottfriedia* sp. OAE603.

The inoculums were distributed to five participants: Lawrence Berkeley National Laboratory (LBNL), USA (organizer); the University of Melbourne, Australia; the University of North Carolina at Chapel Hill, USA; Forschungszentrum Jülich, Germany; and the Max Planck Institute for Plant Breeding Research, Germany. Empty tubes were shipped first to estimate speed, dry ice sublimation, and select vendors. Shipments included content declaration, receiver cover letter, and country-specific permits. Identifying the closest phylogenetic species for each isolate helped avoid customs delays. Shipments to Germany were classified as biosafety level 1 (BSL1) and sent via FedEx International Priority with 8.2 kg of dry ice as a delivered-at-place with a proforma invoice detailing freight charges and goods value. The SynCom import to Australia required a permit for conditionally non-prohibited goods under the Biosecurity Act 2015 Section 179 (1) and was sent via Aeronet Worldwide with 22.68 kg of dry ice, which was refilled during the 9-day transit. The shipment to North Carolina used FedEx Standard Overnight with 9 kg of dry ice.

### 4.2. Experimental setup and plant growth conditions

Participating laboratories assembled EcoFAB 2.0 devices and followed the experimental protocol (https://dx.doi.org/10.17504/protocols.io.kxygxyydkl8j/v1) [15]. In summary, *B. distachyon* Bd21-3 seeds were surface-sterilized, plated on ½ MS basal salts with 1.5% phytoagar, and stratified for 3 days at 4 °C. The plates were then moved to the growth chamber with a 14 h photoperiod (16 h for lab C) at 26 °C and 10 h dark at 20 °C with photosynthetic photon flux density (PPFD) at 110–140 µmol/m²/s and 70% humidity if tunable. Each lab used two data loggers (HOBO MX2202 and UA-002-64) to track illuminance in lux normalized to maximum value per lab (for dark duration confirmation) and temperature. After 3 days, germinated seeds were transferred to sterile EcoFAB 2.0 devices with 9 mL of ½ MS basal salts (pH 5.5–6.0) and placed back in the chamber. After 4 days, plants were inoculated with Mock, SynCom16, or SynCom17, resuspended from 100 µL stocks to 10 ml using sterile ½ MS basal salts. Each EcoFAB 2.0 device received 1 ml of inoculum, resulting in a glycerol concentration of 0.02% in 10 ml of hydroponic medium. The treatment groups included: i) Mock-inoculated axenic plant control, ii) SynCom16-inoculated plants, iii) SynCom17-inoculated plants, and iv) Mock-inoculated technical control (plant-free). The sterility of the uninoculated devices was tested at 0 and 22 DAI by incubating hydroponic medium on LB agar plates for 22 days. Evaporated water was re-supplied and roots imaged by scanning every seven days. Some laboratories shifted their root imaging timepoints based on scanner availability. Plant harvest and sampling happened at 22 DAI.

### 4.3. Root phenotyping

To automate root analysis from flatbed scanner images, we used RhizoNet, a workflow for precise root segmentation, ideal for tangled roots in EcoFAB 2.0 devices [38]. Lab A's scans faced issues with condensation and reflections, causing

low contrast; thus, SmartRoot V4.21 was used instead [39]. The raw root measurements from both methods were normalized to the maximum value for each lab.

### 4.4. Samples collection and shipment

Sample collection and initial processing were performed independently in each lab. We collected 50 μL of the unfiltered growth medium for amplicon sequencing. The remaining spent medium was filtered via a 0.2 μm PES filter for metabolomic analysis, collected in polypropylene tubes (lab A: VWR 21008-103, labs B–E: VWR 93000-026) and stored at −80 °C before shipment. The root and shoot were separated during plant harvest, fresh weights were measured, roots were frozen for microbiome analysis by amplicon sequencing, and shoots were lyophilized for dry weight measurement. The filtered medium, unfiltered medium, and frozen roots were shipped to LBNL on dry ice with gel packs. The loggers were shipped separately at room temperature. To import intact frozen *B. distachyon* roots to the USA required a Controlled Import Permit (PPQ Form 588), supplier declaration, and TSCA Certification. Samples from Germany were shipped via DHL Medical Express (2 days) on 10 kg of dry ice, from Australia via Cryopdp (5 days) on 24 kg dry ice with a refill request, and from North Carolina via FedEx Priority Overnight on 9 kg dry ice (1 day).

### 4.5. Analysis of rhizosphere metabolites

At the LBNL, hydroponic medium samples were lyophilized (Labconco, Kansas City, MO). Empty tubes were used as extraction controls. During extraction, samples were kept on dry ice, and Solvents were chilled at −20 °C. The dried material was suspended in 1 mL methanol (MX0486, Sigma), vortexed, and transferred to a microcentrifuge tube, followed by residue collection with an additional 0.5 mL methanol. Samples were then sonicated (97043-944, VWR) in ice water for 15 min, centrifuged at $10,000g$ for 5 min at 10 °C, and supernatants transferred to new tubes. Supernatants were dried overnight by vacuum concentration (SpeedVac, Thermo). The following morning, samples were resuspended in 150 μL methanol containing internal standard mix (S5 Table), vortexed, and centrifuged at 10,000g for 5 min at 10 °C. Supernatants were filtered with 0.22 μm PVDF filters (UFC30GV, Millipore), transferred to amber glass vials with inserts (5,188−6,592, Agilent), and sealed with screw caps (5,185−5,820, Agilent). Samples were analyzed by LC-MS/MS, with polar metabolites separation using hydrophilic liquid interaction chromatography (column 683775-924, Agilent) on an Agilent 1290 HPLC system, followed by detection in positive ion mode on a Thermo Orbitrap Exploris 120 Mass Spectrometer (S5 Table). Samples were injected in positive mode, with methanol blanks between each sample; internal and external controls were used for quality control.

For untargeted metabolomics, the mzML files were processed via MZMine 3.0 [40] to create a feature list using a custom batch process (S1 File). The features with MS2 spectrum were then annotated in GNPS2 using spectral metabolite libraries [41]. Features with MS2, retention time (RT) > 0.6 min, and exudate sample max peak height > 10× of extraction and technical controls were included, resulting in 833 features across all samples and labs. For targeted metabolomics, metabolites were identified (S6 Table) by analyzing the data with an in-house library of $m/z$, RT, and MS2 fragmentation information from authentic reference standards using Metabolite Atlas (https://github.com/biorack/metatlas) [42]. Only metabolites with a maximum exudate sample peak height > 3× of extraction and technical controls were included. The identified metabolites were manually classified using the PubChem Classification Browser (https://pubchem.ncbi.nlm.nih.gov/classification/).

### 4.6. Microbiome analysis by amplicon sequencing

DNA was extracted from ground roots and media using the DNeasy PowerSoil Pro Kit (Qiagen), following the manufacturer's instructions with minor modifications. Ground roots were mixed with a resuspension buffer, transferred to bead-beating tubes, and frozen at −80 °C. Samples were then thawed at 60 °C and bead-beaten using the FastPrep-24 system for 30 s at setting 5.0 (MP Biomedicals). The elution buffer pre-heated to 60 °C. Samples were extracted

in batches with water-only samples as negative process controls. Glycerol stocks for 16- and 17-SynCom and a glycerol-only mock were included for time-zero data.

PCR amplification of V4 amplicons was performed in two steps. Library amplicons were generated using the Illumina i7 and i5 index/adapter sequences with V4 primers 515F (Parada) (GTGYCAGCMGCCGCGGTAA) and 806R (Apprill) (GGACTACNVGGGTWTCTAAT) [43]. Amplification used pooled primers on a Bio-RAD CFX 384 Real-Time PCR system with QuantiNova SYBR Green PCR kit (Qiagen) in 10 µL reactions with primers at 4 µM and mitochondrial and chloroplast PNA blockers (PNA Bio) at 1.25 µM. Amplification was initiated at 95 °C for 3 min, followed by the cycle: 95 °C for 8 s, 78 °C for 10 s, 54 °C for 5 s, and 60 °C for 30 s, followed by fluorescence measurement. Root samples were amplified for 22 cycles and media samples for 30 cycles with at least 1 replicate. To allow direct comparison, the SynCom16 and SynCom17 inocula were each amplified for both 22 and 30 PCR cycles, mirroring the amplification conditions used for root and media samples, respectively. Libraries were purified at least twice to remove excess primers using the Mag-Bind TotalPure NGS beads (0.8×), and targets were quantified using the QuantiFluor dsDNA System (Promega). Libraries were then pooled (i.e., root and media samples separately) and purified again. The concentration of the pooled library was determined using the Qubit dsDNA Assay Kit (BR or HS, Invitrogen). The flowcell was sequenced on the Illumina MiSeq sequencer using MiSeq Reagent kits, V3 (600-cycle), following a 2 × 00 indexed run recipe. MiSeq reads were processed using Usearch (v11.0.667) [44]. Reads were merged, trimmed, and short sequences removed using the 'fastq_mergepairs, fastx_truncate, and fastq_filter' commands. An initial OTU table was generated with the 'fastx_uniques, cluster_otus, and otutab' functions, and OTUs were assigned to SynCom members using the 'annot' function with V4 reference sequences. The three samples with the most unknown reads were analyzed via NCBI Microbial Nucleotide BLAST with Representative Genomes database and MegaBLAST algorithm to identify potential matches. The relative abundance of each SynCom member was calculated as the proportion of total microbial reads (excluding plant reads) per sample.

### 4.7. Growth and biofilm formation assays for bacterial isolates

Lab A conducted the biofilm formation assays. The crystal violet assay for biofilm formation was modified from a previous method [45]. Isolates were grown in R2A, washed, and resuspended in a 30 mM phosphate buffer. They were inoculated into the plates with NLDM medium [20] at a 1:10 (v/v) ratio (final volume of 100 µL and initial OD$_{600}$ of 0.02) and incubated statically at 30 °C for 3 days ($n = 4$–5) and growth was measured at OD$_{600}$. After incubation, wells were washed 3× with MilliQ water, air-dried, and stained with 125 µL of crystal violet solution (0.1% v/v crystal violet, 1% v/v methanol, and 1% v/v isopropanol in Milli-Q water) for 30 min at RT. Wells were rinsed 3× with MilliQ water, destained with 125 µL of 30% acetic acid for 30–60 min at RT, and absorbance at 550 nm (OD$_{550}$) of the destaining solution was measured.

### 4.8. Genomic analysis of bacterial isolates

All isolates were grown in R2A except *Bradyrhizobium* sp. OAE829, which was grown in 1/10 R2A. High molecular weight genomic DNA was extracted from bacterial pellets with the Monarch HMW DNA Extraction Kit (New England Biolabs) or the MasterPure Complete DNA Purification Kit (Lucigen). The genomic DNA was submitted to the JGI for sequencing using PacBio Sequel II or Illumina NovaSeq S4. Genomes were stored in the Genomes OnLine Database (GOLD) [46], followed by submission to the Integrated Microbial Genomes and Microbiomes (IMG/M) (https://img.jgi.doe.gov/) for annotation [47]. The annotated genomes can be accessed via Hugging Face (https://doi.org/10.57967/hf/5885) [16] or IMG/M under the listed Taxon ID or GOLD Project ID (S2 Table): *Arthrobacter* sp. OAP107 (2931867202, Gp0588953), *Gottfriedia* sp. OAE603 (2931797537, Gp0588949, formerly known as *Bacillus* sp. OAE603 [11]), *Bosea* sp. OAE506 (2931782253, Gp0589672), *Bradyrhizobium* sp. OAE829 (2931808876, Gp0589676), *Brevibacillus* sp. OAP136 (2931855177, Gp0588951), *Paraburkholderia* sp. OAS925 (2931840637, Gp0589681, formerly known as *Burkholderia* sp. OAS925 [9]), *Chitinophaga* sp. OAE865 (2931817136, Gp0589677), *Lysobacter* sp. OAE881 (2931823763, Gp0589678), *Marmoricola*

sp. OAE513 (2931787146, Gp0589673), *Methylobacterium* sp. OAE515 (2931791092, Gp0589674), *Mucilaginibacter* sp. OAE612 (2931861231, Gp0588952), *Mycobacterium* sp. OAE908 (2852593896, Gp0440934), *Niastella* sp. OAS944 (2931847253, Gp0589682), *Paenibacillus* sp. OAE614 (2931801854, Gp0589675), *Rhizobium* sp. OAE497 (2931775946, Gp0589671), *Rhodococcus* sp. OAS809 (2931833612, Gp0589680), *Variovorax* sp. OAS795 (2931827682, Gp0588950).

Comparative genomics was conducted using genome statistics and KEGG pathways protein-coding gene abundance from IMG/M. Based on the genes involved in acid resistance of *Escherichia coli*, including Glutamine/Glutamate and Arginine membrane transporters (GadC and AdiC), glutaminase (YbaS), and decarboxylases (GadA, GadB, and AdiA) [29], we searched for genes annotated with similar functions (GltIJKL, HisPMQ-ArgT, GlnHPQ, glsA, GAD, AdiA). KEGG module coverage was compared between *Paraburkholderia* sp. OAS925 and the other 16 genomes using the "Statistical Analysis" tool from IMG [48] using Fisher's exact test and modules with a corrected $p$-value $<10^{-05}$ were manually inspected for a potential link to plant root colonization.

A phylogenomic tree of the 17 SynCom members was constructed using the GTDB-tk workflow [49], incorporating 120 marker proteins for multiple sequence alignment. Fasttree [50] generated the tree, which included two closely related genomes for context, and it was visualized with Interactive Tree Of Life (iTOL) [51]. The phylum-level taxonomic classification is indicated for each member. Bootstrap values, derived from 100 replicates, are displayed for nodes with over 50 bootstrap support values. *Chloroflexus aggregans* served as an outgroup to root the tree.

### 4.9. Fluorescent microscopy

Lab A conducted the plant growth and microscopy. The pGinger plasmid 23100 containing the RFP gene under the kanamycin (Kan) resistance marker was introduced into *Paraburkholderia* sp. OAS925, as described previously [52]. Briefly, 1 mL of *Paraburkholderia* sp. OAS925 grown overnight at 30 °C in R2A medium was mixed with 1 mL of *E. coli* S17 dapE-harboring the pGinger plasmid grown overnight at 37 °C on LB medium with Kan at 50 µg/mL and diaminopimelic acid (DAP) at 300 µM. The mixture was pelleted for 1 min at 10,000$g$ and then resuspended in 100 µL water with 300 µM DAP. This mixture was then placed onto an R2A agar plate and incubated overnight at 30 °C. The bacterial mix was then scraped, resuspended in water, and plated on R2A with Kan 50 µg/mL. Transconjugants were verified via fluorescent microscopy and colony PCR.

The *Paraburkholderia* sp. OAS925 expressing RFP was grown overnight in 7 mL of R2A medium with Kan 20 mg/L, shaken at 200 rpm at 27 °C. The cells were pelleted by centrifugation at 4,000$g$ for 10 min, resuspended in ½ MS basal salts, and used to inoculate 3-day-old plants in EcoFABs 2.0 at a starting $OD_{600}$ of 0.01. A flatbed scanner captured root architecture to indicate microscopy locations at 1 and 3 DAI. The EVOS M5000 imaging system (Thermo Fisher) was used for inverted microscopy, merging txRED and bright-field images. Uninoculated plants served as controls for autofluorescence.

### 4.10. Motility assays

Lab A conducted the motility assays. Pre-cultures were grown in 8 ml of liquid medium shaken at 200 rpm, at 27 °C in the dark, for 5–8 days. The swimming motility was tested by observing colony spreading on R2A soft agar plates (0.3% w/v). Initially, motility was tested for all 17 isolates at pH 7.2. Then, *Paraburkholderia*, *Gottfriedia*, or *Brevibacillus* were tested at pH 4, 5, 6, 7, 8, or 9 (adjusted with 1M HCl or 0.5M NaOH). Each 10 cm Petri dish containing 30 ml of solidified medium was inoculated with 5 µL of culture at the center ($n = 3$). Plates were incubated at 27 °C in the dark, and the motility ring diameter was measured after 24 and 45 h.

### 4.11. Data management, statistical analyses, software, and data visualization

The participating laboratories uploaded plant biomass data, root scans, and photos into a shared Google folder with structured directories and spreadsheets. The heat maps were generated with RStudio version 4.0.5 using heatmap.2 in the ggplots package [53]. The NMDS plots were also generated in RStudio using vegan package version 2.5-7

[54]. GraphPad Prism 10 version 10.2.3 generated all other plots and statistical analyses. Biorender.com was used to create graphical overviews. Microsoft Excel version 16.78.3 was used to store and manipulate data frames.

Statistical analysis of cross-laboratory variation included a two-way ANOVA by measuring significance of main effects and their interactions, calculation of source of variation as sum of squares of main effects and total sum of squares, and CV that was measured as a ratio between the standard deviation $\sigma$ to the mean $\mu$. The statistical analysis was applied to microbe abundance data from 16S rRNA sequencing and raw peak heights for 60 identified rhizosphere metabolites. Additionally, metabolite intensities were also analyzed by pairwise comparisons. The ANOVA and pairwise tests were done in Python v 3.8 using the Pingouin package version 0.5.5. The default ANOVA command was used between lab and treatment. The default pairwise tests command was used between lab and treatment with a $p$-value adjustment of Benjamini/Hochberg FDR correction. The code can be found at https://doi.org/10.6084/m9.figshare.29700029.

## Supporting information

**S1 Fig. Plant biomass data for laboratories A–E.** Box plots display all data points, with hinges spanning the 25th to 75th percentiles, a central line denoting the median, and whiskers reaching the minimum and maximum values. Different lowercase letters indicate statistically significant differences at $p < 0.05$. One-way ANOVA with Tukey test ($n = 7$). The shoot photos can be found at https://doi.org/10.6084/m9.figshare.26409310. The data underlying this figure can be found at https://doi.org/10.6084/m9.figshare.26401315.
(TIFF)

**S2 Fig. Plant growth conditions in labs A–E.** (a) Temperature ($T$) and (b) Normalized illuminance (% of lab maximum value) to assess lights-off duration, measured by HOBO loggers (Pendant model #UA-00264 in red, Bluetooth model #MX2202 in blue). Dashed lines show the set day/night $T$ (26/20 °C); gray areas mark the 7-day pre-inoculation period. Labs A, B, and E experienced logging interruptions due to battery drainage; Lab D's Bluetooth logger did not cover the experimental period. The data underlying this figure can be found at https://doi.org/10.6084/m9.figshare.26401315.
(TIFF)

**S3 Fig. Representative images of root systems.** Labs A–E imaged roots in EcoFAB 2.0 devices with flatbed scanners. The figure shows plants at harvest (22 DAI). The root scans and Rhizonet reports can be found at https://doi.org/10.6084/m9.figshare.26131291. The data underlying this figure can be found at https://doi.org/10.6084/m9.figshare.26401315.
(TIFF)

**S4 Fig. Microbiome composition.** (a) Microbiome in plant growth media and starting inoculum. Letters indicate different laboratories, with each biological replicate shown ($n = 7$). The inoculum shows technical replicates ($n = 3$). **(b)** NMDS plot of root and media microbiomes with 95% confidence ellipse. Different laboratories are shown with various symbols, while colors represent SynCom16 (blue) versus SynCom17 (orange) inoculated plants. The data underlying this figure can be found at https://doi.org/10.6084/m9.figshare.26401315 or in S3 Table.
(TIFF)

**S5 Fig. Untargeted metabolomics on rhizosphere metabolites.** NMDS plots with a 95% confidence ellipse for 833 filter features for individual laboratories A–E and all combined. Different colors show treatments: Axenic (gray), SynCom16 (blue), and SynCom17 (orange), while shapes indicate laboratories in the combined plot. The data underlying this figure can be found at https://doi.org/10.6084/m9.figshare.26401315.
(TIFF)

**S6 Fig. Statistical analysis of 16S rRNA sequencing data. (a)** ANOVA $p$-value of effects for root and media. Each point is a SynCom member with blue indicating significant values (red line $p = 0.05$). **(b)** ANOVA source of variation for root and media calculated from ratio of squares (SS) for each effect and total sum of squares (TSS). **(c)** Coefficient of variation

(CV = $\sigma/\mu$) distribution (median in red) for treatments within (blue) and across (gray) labs for root and media. The data underlying this figure can be found at https://doi.org/10.6084/m9.figshare.26401315.
(TIFF)

**S7 Fig. Statistical analysis of 60 identified rhizosphere metabolites. (a)** ANOVA $p$-value of effects for metabolite peak heights. Each point is a metabolite with blue indicating significant values (red line $p = 0.05$). **(b)** Number of statistically different metabolites ($p < 0.05$) in lab pairwise comparisons. **(c)** ANOVA source of variation for root and media calculated from ratio of squares (SS) for each effect and total sum of squares (TSS). **(d)** Coefficient of variation (CV = $\sigma/\mu$) distribution (median in red) for treatments within (blue) and across (gray) labs. The data underlying this figure can be found at https://doi.org/10.6084/m9.figshare.26401315.
(TIFF)

**S8 Fig. Comparative genomics of bacterial isolates.** Bar graphs show genome characteristics (from left: the abundance of bases, coding, G + C bases, total genes, and KEGG pathway genes. The search for acid resistance system genes included GAD (EC 4.1.1.15 glutamate decarboxylase), AdiA (EC 4.1.1.19 arginine decarboxylase), glsA (EC 3.5.1.2 glutaminase), GlnHPQ (glutamine ABC transporter), GltIJKL (glutamate/aspartate ABC transporter), HisPMQ-ArgT (arginine/ornithine ABC transporter). The heat map shows normalized gene abundance for selected KEGG pathways. The data underlying this figure can be found at https://doi.org/10.6084/m9.figshare.26401315.
(TIFF)

**S9 Fig. Fluorescent microscopy in EcoFAB 2.0 conducted on plants grown in the lab A.** We inoculated RFP-expressing *Paraburkholderia* sp. OAS925 into the *B. distachyon* rhizosphere in EcoFAB 2.0 devices on the day of transfer (DAT) for the seedling. The plots show **(a)** uninoculated plant control (1 DAT) and medium-inoculated plants ($OD_{600}$ 0.01) at **(b)** 1 and **(c)** 3 days after inoculation (DAI). EcoFAB 2.0 root scans indicate the locations for microscopy. The inset microscopy images show merged TxRed and bright-field (BF) channels. The microscopy images can be found at https://doi.org/10.6084/m9.figshare.26449852.
(TIFF)

**S10 Fig. Assessment of growth and biofilm formation for bacterial isolates in vitro by the lab A.** (Top) Biofilm was measured with crystal violet (CV) staining at $OD_{550}$, and (Bottom) isolate growth was assessed based on $OD_{600}$ values of isolates on the defined NLDM liquid medium for 3 days. The horizontal dotted line indicates the mean value of sterile medium negative control (NC). Different letters indicate statistically significant differences at $p < 0.05$, One-way ANOVA with Tukey's test, $n = 4$–5 for isolates and $n = 11$ for NC. The data underlying this figure can be found at https://doi.org/10.6084/m9.figshare.26401315.
(TIFF)

**S11 Fig. Motility assays in the lab A. (a)** The initial screen for swimming motility across bacterial isolates was measured 24 and 45 h after inoculation. **(b)** Phenotypes of the most motile strains at 45 h since inoculation. **(c)** pH effects on the motility ring diameter of isolates with bulls-eye colony morphology. Images of motility assays with bacterial isolates can be found at https://doi.org/10.6084/m9.figshare.26457928. The data underlying this figure can be found at https://doi.org/10.6084/m9.figshare.26401315.
(TIFF)

**S1 Table. Chamber settings and models and experiment timing.**
(XLSX)

**S2 Table. SynCom members overview, genome information, and their $OD_{600}$ to CFU conversion ratios.**
(XLSX)

**S3 Table. 16S rRNA sequencing read counts in media and root samples.** Samples labeled "none" are Axenic plant controls, "Gly" are glycerol stocks of inoculum.
(XLSX)

**S4 Table. BLAST search results for unknown reads in selected media samples.**
(XLSX)

**S5 Table. LC–MS/MS parameters.**
(XLSX)

**S6 Table. Metabolite identification and intensity.** Sample names follow code S-L-T, where S is sample (ExCtrl = extraction control, RtExu = root exudate sample, TxCtrl = technical control); L is the laboratory (A–E letters for each lab); T is the treatment (Axenic, SynCom16 or SynCom17). The prefix PkHt = Peak height.
(XLSX)

**S7 Table. ANOVA results for 16S rRNA sequencing data and metabolite intensity.**
(XLSX)

**S8 Table. Pairwise results for metabolites.**
(XLSX)

**S9 Table. Comparison of KEGG modules between SynCom16 vs. *Paraburkholderia* sp. OAS925.**
(XLSX)

**S1 Video. Fluorescent microscopy at 1 DAI.** Z-Stack shows RFP-*Paraburkholderia* motility and colonization on *B. distachyon* roots at 1 DAI in EcoFAB 2.0.
(MP4)

**S2 Video. Fluorescent microscopy at 3 DAI.** Root colonization by RFP-*Paraburkholderia* at 3 DAI in EcoFAB 2.0. Merged bright-field and TxRed channels followed by TxRed footage.
(MP4)

**S1 File. MZMine 3 settings.**
(ZIP)

## Acknowledgments

We would like to thank Julio Corral and Kaosio Saephan for coordinating the shipment and receipt of samples and Chips Hoai for guiding us regarding regulations related to the import and export of samples. We thank Diana Dresbach for technical help with the trial at the Max Planck Institute.

Data analysis utilized resources from the National Energy Research Scientific Computing Center, a DOE Office of Science User Facility (Contract No. DE-AC02-05CH11231). A portion of these data was produced by the US Department of Energy Joint Genome Institute (https://ror.org/04xm1d337; operated under Contract No. DE-AC02-05CH11231) as part of projects 10.46936/10.25585/60001370 and 10.46936/10.25585/60001258.

## Author contributions

**Conceptualization:** Vlastimil Novak, Peter F. Andeer, Karsten Zengler, John P. Vogel, Trent R. Northen.

**Data curation:** Vlastimil Novak, Peter F. Andeer, Eoghan King, Jacob Calabria, Connor Fitzpatrick, Jana M. Kelm, Kathrin Wippel, Zineb Sordo, Simon Roux, Daniela Ushizima, Borjana Arsova, Jeffery L. Dangl, Paul Schulze-Lefert, Michelle Watt, John P. Vogel, Trent R. Northen.

**Formal analysis:** Vlastimil Novak, Peter F. Andeer, Eoghan King, Suzanne M. Kosina, Benjamin P. Bowen, Archana Yadav, Zineb Sordo, Simon Roux, Adam M. Deutschbauer, Daniela Ushizima, John P. Vogel, Trent R. Northen.

**Funding acquisition:** Romy Chakraborty, Adam M. Deutschbauer, Karsten Zengler, John P. Vogel, Trent R. Northen.

**Investigation:** Vlastimil Novak, Peter F. Andeer, Eoghan King, Jacob Calabria, Connor Fitzpatrick, Jana M. Kelm, Kathrin Wippel, Chris Daum, Matthew Zane, Mingfei Chen, Dor Russ, Trenton K. Owens, Yezhang Ding, John P. Vogel, Trent R. Northen.

**Methodology:** Vlastimil Novak, Peter F. Andeer, Eoghan King, Bradie Lee, John P. Vogel, Trent R. Northen.

**Project administration:** Vlastimil Novak, John P. Vogel, Trent R. Northen.

**Resources:** Vlastimil Novak, Chris Daum, Matthew Zane, Catharine A. Adams, Romy Chakraborty, Michelle Watt, John P. Vogel, Trent R. Northen.

**Software:** Vlastimil Novak.

**Supervision:** Romy Chakraborty, Simon Roux, Adam M. Deutschbauer, Daniela Ushizima, Karsten Zengler, Borjana Arsova, Jeffery L. Dangl, Paul Schulze-Lefert, Michelle Watt, John P. Vogel, Trent R. Northen.

**Validation:** Vlastimil Novak, Suzanne M. Kosina, John P. Vogel, Trent R. Northen.

**Visualization:** Vlastimil Novak, Archana Yadav, John P. Vogel, Trent R. Northen.

**Writing – original draft:** Vlastimil Novak, Peter F. Andeer, John P. Vogel, Trent R. Northen.

**Writing – review & editing:** Vlastimil Novak, Peter F. Andeer, Eoghan King, Jacob Calabria, Connor Fitzpatrick, Jana M. Kelm, Kathrin Wippel, Suzanne M. Kosina, Benjamin P. Bowen, Chris Daum, Matthew Zane, Archana Yadav, Mingfei Chen, Dor Russ, Catharine A. Adams, Trenton K. Owens, Bradie Lee, Yezhang Ding, Zineb Sordo, Romy Chakraborty, Simon Roux, Adam M. Deutschbauer, Daniela Ushizima, Karsten Zengler, Borjana Arsova, Jeffery L. Dangl, Paul Schulze-Lefert, Michelle Watt, John P. Vogel, Trent R. Northen.

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
