## [Editor Report · Decision Letter 0]

17 Jan 2025

Dear Trent,

Thank you for submitting your manuscript entitled "Multi-laboratory Study Establishes Reproducible Methods for Plant-Microbiome Research in Fabricated Ecosystems" for consideration as a Methods and Resources by PLOS Biology.

Your manuscript has now been evaluated by the PLOS Biology editorial staff, as well as by an academic editor with relevant expertise, and I am writing to let you know that we would like to send your submission out for external peer review. We would like to suggest to change the type of article to Meta-Research Article since we think might fit better. Meta-Research Articles examine how biological research is designed, carried out, communicated and evaluated; We welcome both exploratory and confirmatory research that has the potential to drive change in research and evaluation practices in the life sciences and beyond.

However, before we can send your manuscript to reviewers, we need you to complete your submission by providing the metadata that is required for full assessment. To this end, please login to Editorial Manager where you will find the paper in the 'Submissions Needing Revisions' folder on your homepage. Please click 'Revise Submission' from the Action Links and complete all additional questions in the submission questionnaire. If you agree with the change in Article type, please, when adding the rest of the metadata choose "Meta-Research Article".

Once your full submission is complete, your paper will undergo a series of checks in preparation for peer review. After your manuscript has passed the checks it will be sent out for review. To provide the metadata for your submission, please Login to Editorial Manager (https://www.editorialmanager.com/pbiology) within two working days, i.e. by Jan 19 2025 11:59PM.

Have a nice weekend,

Melissa

Melissa Vazquez Hernandez, Ph.D.

Associate Editor

PLOS Biology

---

## [Decision Letter · Decision Letter 1]

26 Mar 2025

Dear Trent,

Thank you for your patience while your manuscript "Multi-laboratory Study Establishes Reproducible Methods for Plant-Microbiome Research in Fabricated Ecosystems" was peer-reviewed at PLOS Biology. It has now been evaluated by the PLOS Biology editors, an Academic Editor with relevant expertise, and by two independent reviewers. First of all I would like to apologize for the extremely long delay on giving you a decision, this is really out of the normal.

In light of the reviews, which you will find at the end of this email, we would like to invite you to revise the work to thoroughly address the reviewers' reports. As you will see below, the reviewers are positive about the relevance and novelty of the study, yet some concerns have raised during revision. Reviewer 1 mentions that you might be missing additional analysis on the differences that exist between the labs. The reviewer suggests to model microbiome data including the lab as a factor, consider how uneven the inoculum seems to be, and clarify several points on their experiments. While most of the comments from Reviewer 2 could be addressed with further discussion and clarification in the text, the reviewer mentions that perhaps you should re-do the experiments to ensure absence of contamination. Additionally, the reviewer mentions some experiments to answer e.g. how does a single strain Paraburkholderia sp. OAS925 affects B. distachyon growth.

IMPORTANT: after discussion with the Academic Editor and the reviewers, while we think reanalysis of the data might be necessary, we do not expect that you re-do any experiment as contamination tends to be a reality in such studies.

Given the extent of revision needed, we cannot make a decision about publication until we have seen the revised manuscript and your response to the reviewers' comments. Your revised manuscript is likely to be sent for further evaluation by all or a subset of the reviewers.

**IMPORTANT - SUBMITTING YOUR REVISION**

*Re-submission Checklist*

*Published Peer Review*

*PLOS Data Policy*

*Blot and Gel Data Policy*

Sincerely,

Melissa

Melissa Vazquez Hernandez, Ph.D.

Associate Editor

PLOS Biology

REVIEWERS' COMMENTS:

Reviewer #1:

This manuscript addresses the important and rarely-investigated issue of replicability of results between labs, specifically applied here to microbiome research. The organizers got together 5 participating labs and distributed an axenic growth chamber, pre-mixed microbial communities, plant seeds, and specific instructions to attempt to reproduce the same experiment as faithfully as possible. Much, but not all, of the experiment replicated. The paper is interesting, but should do some additional analyses on the results, particularly related to the differences that do exist between labs. The authors focused mainly on how Paraburkholderia completely overtook the community in all labs. To some extent, the detailed look at Paraburkholderia feels like a bit of a distraction from the main interesting points about reproducibility, because it is SOOO dominant. Focusing on this obscures other instructive differences, which I think should be more central. More specific comments follow, not in any particular order.

>> The authors should model microbiome data including lab as a factor. How much of the variation in community in SynCom 16 is due to lab? How much remains unexplained? For community 17, after removing Paraburkholderia reads, what is left? There are unsatisfyingly few stats.

>> A couple things are curious about the inoculum. First, it appears quite uneven - I suppose the 16 or 17 members were intended to be mixed quite evenly. Can this be further explained? Second, the inoculum seems to bear nearly no resemblance to the ultimate community. Even Paraburkholderia seems not to be present in any substantial amount in the inoculum. Can the authors please present a table of percentages (in supplemental would be OK) that lists the median % of each community member in the inoculum and also in the ultimate community?

>>Related to the inoculum, it is odd that the inoculum in Figure 3 (plants) appears different than the inoculum in figure S3 (media). Were the media and plant inoculations NOT done simultaneously, such that independent inocula batches were thawed for the media?

>>Particularly the media in Fig. S3, there are visible bars of "unknown" sequences colored black. What is the diversity of "unknown" in this? For example, if the "unknown" is all the same thing, isn't that a clear case of contamination? And if it's a mix of different things, how to explain it?

>> it is odd that the inoculum in Figure 3 (plants) appears different than the inoculum in figure S3 (media). Were the media and plant inoculations NOT done simultaneously, such that independent inocula batches were thawed for the media?

>> For some analysis (eg microscopy) was done in the main host lab. I think it is worth identifying which of the labs that was. Those analysis could mention, for example, that microscropy was conducted on plants grown in Lab A.

>> In Figure S1, each column (group of measurements of the same phenotype) should have the same Y axis so that the plots can be compared. And I think it's further preferable that columns and rows are switched, such that one can look across from left to right and see the shoot dry weight in each lab, for example.

>>Likewise in Figure S2, the Y axes should be made comparable. The temperature side is OK, but the light levels are highly variable across labs and this should be immediately apparent looking at the graphs, but it's somewhat hidden due to the variable axes.

>>Fig. 2A is confusing. The barplots look like this is a continuous distribution, but as I understand it, if a device was sterile, it gets a grey box, and if it's contaminated, it gets a black box. It would be clearer if the boxes were separated so they could be seen as discrete data points, or even I think it would be clearer if the Y axis showed # of contaminated boxes and most are 0 and those that got contaminated go to 1. For example. But it's confusing as is.

>> in Fig 2C, there are some differences between the labs in terms of measurement times. Lab B took a measurement at 4 days? Lab E did 0 days but skipped 14 days? What happened here, a miscommunication or technical failure or something?

>> Figure S4, putting each lab with a totally different NMDS2 plot is difficult (for me) to interpret. Is not it easier and more informative to simply use the combined lab plot to generate the spacings for the NMDS and simply erase the points from the other labs so that points from a single lab can be shown individually?

>> In the methods, please include the min/max/median sequencing depth for the microbiome samples.

>>Line 276.. the authors say the ecofab devices are "free". Surely this can't be true, or must have some limits. Please revise to clarify. It means free if one is part of a JGI user program? Is that limited to USA? Grant recipients of some form? Etc.

Reviewer #2 (Mengcen Wang):

This study addresses a critical challenge in microbiome research, inter-laboratory reproducibility, by assessing the consistency of synthetic community assembly experiments across multiple laboratories. The authors leverage ecosystems involving Brachypodium distachyon, two different synthetic bacterial communities, and sterile EcoFAB 2.0 devices, providing valuable insights into inoculum-dependent microbiome shifts and plant responses. While the study is generally well-structured and presents important findings, some areas require further clarification and improvement:

1. It was indicated that the root system development was analyzed using RhizoNet (Lab B-E) and ImageJ (Lab A). Why did these five laboratories not standardize their analytical methods to minimize variability and improve reproducibility? Furthermore, it would be beneficial to include representative images of root systems.

2. In line135-139, the uninoculated EcoFAB 2.0 sterility tests across laboratories A-E shown that the treatments of laboratory D in SynCom17 and laboratory B in medium-only control were contaminated. It would be more appropriate to redo the experiments to ensure the absence of contamination and further validate the reliability of the results.

3. The SynCom17, which contained the dominating bacteria Paraburkholderia sp. OAS925 could reduce root growth of B. distachyon. How about the effects of the single strain Paraburkholderia sp. OAS925 on B. distachyon growth?

4. L209-210 what does " its potential to outgrow competitors when cultured with common soil metabolites in the NLDM " mean?

5. L226-L228, "This finding is consistent with a previous study showing Paraburkholderia sp. OAS925 dominance in the B. distachyon root and rhizosphere microbiota and decreased fresh root biomass." However, in Figure 2b, the shoot fresh weight (FW) of plants treated with both SynCom 16 and SynCom 17 was significantly reduced compared to axenic conditions. Moreover, no significant difference was observed between SynCom 16 and SynCom 17 treatments, suggesting that the SynCom 16 community, even without the addition of Paraburkholderia sp. OAS925, is sufficient to reduce shoot FW levels. Therefore, this does not conclusively demonstrate that the reduction in shoot FW under SynCom 17 treatment is specifically driven by Paraburkholderia sp. OAS925.

6. In L303, the manuscript does not clearly explain the rationale behind the 1:1 ratio of strains in the SynComs. It is recommended to provide additional explanation to enhance the scientific rigor and credibility of the study.

7. In L362, the authors indicate that the hydroponic medium sample was used as the root exudate for metabolomic analysis. But how did they rule out potential interference from bacterial metabolites in the medium?

8. The colonization of bacteria in B. distachyon roots can be influenced by various external factors. The assessment of bacterial motility on liquid soft agar only reflects swimming ability in liquid culture media and does not provide definitive evidence that Paraburkholderia sp. OAS925 can outcompete other members during SynCom colonization.

9. The analysis of microbial community and metabolome data is crucial for the research conclusions. However, it is not clear in the paper whether significance analysis was performed on this part of the data and what methods were used. If there is relevant analysis, please provide a complete description.

10. It is recommended to present the analysis conclusions of some data in the supplementary information to enrich the content of the main text.

---

## [Decision Letter · Decision Letter 2]

25 Jul 2025

Dear Trent,

I hope you are doing great. Thank you for your patience while we considered your revised manuscript "Multi-laboratory Study Establishes Reproducible Methods for Plant-Microbiome Research in Fabricated Ecosystems" for publication as a Meta-Research Article at PLOS Biology. This revised version of your manuscript has been evaluated by the PLOS Biology editors, and the original reviewers.

Based on the reviews, we are likely to accept this manuscript for publication, provided you satisfactorily address the remaining editorial points. Please be aware that Academic Editor was unavailable and there might be a possibility that they might have an additional request, which I would communicate with you if it were the case. Please make sure to address the following data and other policy-related requests.

a) We routinely suggest changes to titles to ensure maximum accessibility for a broad, non-specialist readership, and to ensure they reflect the contents of the paper. In this case, we would suggest a minor edit to the title, as follows. Please ensure you change both the manuscript file and the online submission system, as they need to match for final acceptance:

"Protocols, benchmarking datasets and best practices for reproducible research on plant-microbiome interactions"

Please supply the numerical values either in the a supplementary file or as a permanent DOI’d deposition for the following figures:

Figure 2bc, 3, 4, S1, S2ab, S4ab, S5, S6bc, S7bcd, S8, S9, S11ac

*I am aware that there is some raw data in Figshare but it is not clear if it belongs to these figures. If this is the case, I need you to provide the necessary information to be able to replicate each figure. However, it might be easier to just provide an excel file with the raw data for each figure.

c) Please cite the location of the data clearly in all relevant main and supplementary Figure legends, e.g. “The data underlying this Figure can be found in S1 Data” or “The data underlying this Figure can be found in https://doi.org/10.5281/zenodo.XXXXX”

d) Could you please confirm that the repositories you used are publicly available?

e) Please ensure that your Data Statement in the submission system accurately describes where your data can be found and is in final format, as it will be published as written there

We expect to receive your revised manuscript within two weeks.

*Published Peer Review History*

*Press*

Sincerely,

Melissa

Melissa Vazquez Hernandez, Ph.D.

Associate Editor

PLOS Biology

REVIEWERS' COMMENTS:

Reviewer #1: Apologies for taking a couple extra days on the review. After looking at the resubmitted materials, the authors have done a nice job of shifting the focus further towards inter-lab reproducibility, including addressing that statistically. They have also clarified my earlier points of confusion.

The authors have asked "For your convenience below is a figure incorporating the number of amplification cycles to clarify that these reflect the preparation of the different sample types (30 PCR cycles for Media and 22 cycles for Root to achieve similar DNA yield). Let us know if you see a value in including it in the manuscript. We believe that a simple statement in M&M should be sufficient:"

I agree with the authors that the statement they go on to propose is sufficient. I think the paper can be accepted.

Reviewer #2 (Mengcen Wang): The authors have addressed my concerns, and I therefore recommend its publication.

---

## [Editor Report · Decision Letter 3]

10 Aug 2025

Dear Trent,

Thank you for the submission of your revised Meta-Research Article "Breaking the reproducibility barrier with standardized protocols for plant-microbiome research" for publication in PLOS Biology. On behalf of my colleagues and the Academic Editor, Cara Haney, I am pleased to say that we can in principle accept your manuscript for publication, provided you address any remaining formatting and reporting issues. These will be detailed in an email you should receive within 2-3 business days from our colleagues in the journal operations team; no action is required from you until then. Please note that we will not be able to formally accept your manuscript and schedule it for publication until you have completed any requested changes.

PRESS

Sincerely, 

Melissa

Melissa Vazquez Hernandez, Ph.D., Ph.D.

Associate Editor

PLOS Biology
